



# Evolution of DARDAR-CLOUD ice cloud retrievals: new parameters and impacts on the retrieved microphysical properties

Quitterie Cazenave[1,2], Marie Ceccaldi[1], Julien Delanoë[1], Jacques Pelon[3], Silke Groß[2], and Andrew Heymsfield[4]

[1]Université Versailles St-Quentin, LATMOS-IPSL, Guyancourt, France
[2]Institut für Physik der Atmosphäre, Deutsches Zentrum für Luft- und Raumfahrt (DLR), Oberpfaffenhofen, 82234 Weßling, Germany
[3]CNRS/INSU, LATMOS-IPSL, Paris, France
[4]NCAR, Boulder, Colorado, USA

**Correspondence:** Quitterie Cazenave (quitterie.cazenave@latmos.ipsl.fr)

**Abstract.** In this paper we present the latest refinements brought to the DARDAR-CLOUD product, which contains ice cloud microphysical properties retrieved from the cloud radar and lidar measurements from the A-Train mission. Based on a large dataset of in-situ ice cloud measurements collected during several campaigns performed between 2000 and 2007 in different regions of the globe, the parameterizations used in the microphysical model of the algorithm were assessed and refined to

better fit the measurements, keeping the same formalism as proposed in DARDAR basis papers. It is shown that these changes can affect the ice water content retrievals by up to 50%, with, globally, a reduction of the ice water content and ice water path. In parallel, the retrieved effective radii increase between 5% and 40%. The largest differences are found for the warmest temperatures (between $-20\,^{\circ}\mathrm{C}$ and $0\,^{\circ}\mathrm{C}$ ) in regions where the cloud microphysical processes are more complex and where the retrieval is almost exclusively based on radar-only measurements. In regions where lidar measurements are available, the lidar

ratio retrieved for ice clouds is shown to be well constrained by lidar-radar combination or molecular signal detected below thin semi-transparent cirrus. Using this information, the parameterization of the lidar ratio was refined and the new retrieval equals on average 35 sr +/- 10 sr in the temperature range between $-60\,^{\circ}\mathrm{C}$ and $-20\,^{\circ}\mathrm{C}$ .

## 1 Introduction

The impact of ice clouds on the water cycle and radiative budget is still uncertain due to the complexity of cloud processes

that makes it difficult to acquire adequate observations of ice cloud properties and parameterize them into General Circulation Models (GCMs) (Stocker et al., 2013).

Passive and active remote sensing instruments, like visible and infrared (IR) radiometers, cloud radars and lidars, are commonly used to study ice clouds. Inferring cloud microphysical properties like extinction ($\alpha$), ice water content ($IWC$) and effective radius ($r_e$) can be done from one instrument only, or from the synergy of several instruments or channels (i.e. wave-

lengths $\lambda$). Several methods were developed to retrieve ice cloud properties from a single instrument: IR radiometers are commonly used to retrieve integrated $r_e$ from a set of brightness temperatures at different wavelengths (Stubenrauch et al.,





1999; Guignard et al., 2012; Hong et al., 2012), and lidars and radars are useful to retrieve respectively extinction and $IWC$ (Liu and Illingworth, 2000; Vaughan et al., 2004; Heymsfield et al., 2014). However, all of these instruments have shortcomings in different parts of the cloud - for instance, due to the attenuation of the lidar signal, the lidar will be blind in the lower part of a thick cirrus whereas the top of the cloud is invisible to the radar in most cases - resulting in a large spread of values for the retrieved cloud properties. Hence, there is a need to use several instruments to reduce this uncertainty. Synergetic ice properties retrieval methods can combine radiometer with lidar or radar (Evans et al., 2005; Garnier et al., 2012, 2013; Sourdeval et al., 2014) or both lidar and radar (Donovan et al. 2001; Wang and Sassen 2002; Okamoto et al. 2003; Delanoë and Hogan 2008, 2010, hereafter referred to as DH0810).

Radar and lidar are active sensors that provide vertical information on cloud structure and are sensitive to different cloud particle populations. To a first approximation, the radar return signal is proportional to the $6^{th}$ moment of the particle size; hence, within a volume it is most sensitive to the largest particles. On the other hand, lidar backscatter is proportional to the $2^{nd}$ moment of the particle size and is thus more sensitive to particle concentration and backscattering cross-section. Combining the two instruments therefore provides two moments of the particle size distribution. In regions of the cloud where both instruments are available, this method allows a well constrained retrieval of extinction and $IWC$, leading to direct calculation of $r_e$ at each pixel of the vertical profile obtained by this synergy. The difference in sensitivity of the two instruments also gives a more complete view of the cloud structure and microphysics (Donovan et al., 2001; Okamoto et al., 2003; Tinel et al., 2005).

The A-Train constellation of satellites has considerably improved our knowledge of clouds. Since 2006, CALIPSO (Cloud-Aerosol Lidar and Infrared Pathfinder Satellite Observation) and Cloudsat have acquired cloud vertical profiles globally. CALIPSO (Winker et al., 2010) carries CALIOP (Cloud-Aerosol Lidar with Orthogonal Polarization), a lidar operating at 532 nm and 1064 nm with depolarization capabilities on the 532 nm channel (Winker et al., 2007) as well as the Imaging Infrared Radiometer (IIR) and a Wide Field Camera (WFC). CloudSat carries a Cloud Profiling Radar (CPR) measuring reflectivity at 95 GHz (Stephens et al., 2002). Lidar-radar synergetic methods have been adapted to CloudSat and CALIPSO data (Okamoto et al., 2010; Delanoë and Hogan, 2010; Deng et al., 2010). In this paper, we focus on the DARDAR derived products. The DARDAR (raDAR / liDAR) project was initiated by the LATMOS (Laboratoire Atmosphères, Milieux, Observations Spatiales) and the University of Reading. It was developed to retrieve ice cloud properties globally from CloudSat and CALIPSO measurements using a specific universal parameterization of the particle size distribution (Delanoë et al., 2005, 2014) and the Varcloud optimal estimation algorithm (DH0810). DARDAR has 3 products that can be used separately and they are all hosted and available on the Icare (Interactions Clouds Aerosols Radiations Etc) FTP website ftp://ftp.icare.univ-lille1.fr/. The first one is the CS-TRACK product which is the collocated processed A-Train product on the CloudSat track. This product gives the possibility to work on lidar and radar data on the same resolution grid of 1.1 km horizontally and 60 m vertically. From these profiles of active instruments data, a technique for the classification of hydrometeors (called DARDAR-MASK) has been developed. This technique is used to select the lidar-radar range bins (or pixels) where ice cloud properties retrievals (DARDAR-CLOUD) can be performed (Ceccaldi et al., 2013). It is important that the classification is as accurate as possible since including liquid water pixels or noisy pixels in our retrieval could compromise the results. Indeed retrieval techniques are





different for liquid droplets and for ice crystals and a specific analysis should be applied to mixed phase clouds (Hogan et al., 2003). In this paper, we only focus on the retrieval of ice crystal properties.

From collocated profiles of CloudSat and CALIPSO and hydrometeor classification, the DARDAR-CLOUD algorithm performs retrievals of extinction, $IWC$ and $r_e$ at each pixel of ice cloud detection (even when only one instrument is available) on the CS-TRACK grid. The main advantage of DARDAR, compared to many other synergetic methods, is that it seamlessly performs retrievals in cloud regions detected by both the radar and the lidar and in regions detected by only one instrument. This is achieved using an optimal estimation algorithm, finding the best state vector of cloud properties which minimizes the errors on observations (radar reflectivity $Z$ and lidar apparent backscatter $\beta_a$) compared to measurements simulated using a forward model. Whenever one of the measurements is missing, the algorithm relies on an *a priori* estimate of the state vector derived from the climatology.

The DARDAR-CLOUD product has been widely evaluated and used (Deng et al., 2013; Delanoë et al., 2013; Hong and Liu, 2015; Sourdeval et al., 2016; Saito et al., 2017) and a few issued have been identified, such as possible overestimations of $IWC$ (Deng et al., 2013). As a consequence, adjustments have been made on the algorithm to optimize retrievals as a function of range and temperature, especially concerning the detection of ice particles and the cloud microphysical model, keeping the formalism unchanged from DH0810. In the following, the new version of DARDAR-CLOUD resulting from those changes will be called $V_2$ and the version available on the ICARE website until 2018, namely DARDAR-CLOUD v2.1.1, will be referred to as $V_1$. It is important, for the consistency of future studies compared to earlier ones, to give information on the differences between the two versions and the way they impact the results of the algorithm. After introducing the key features of the variational scheme in section 2, its recent updates are detailed in section 3 and their effects on the retrieved cloud microphysical properties are presented in section 4. We will mainly focus on the retrieval of $IWC$ and briefly present the main differences observed on the retrieved particle sizes.

## 2 Key features of DARDAR-CLOUD algorithm

We restate here the main characteristics of the inverse method used for the DARDAR retrievals; readers interested in details of the Varcloud algorithm are invited to check on DH0810.

The method is applied to one profile at a time. We start with a first guess of the state vector on the pixels of the profile where the retrieval can be performed. We select ice only pixels from radar and lidar masks and for the lidar, we restrict these pixels to the ones unobscured by liquid clouds. Indeed lidar signal located below liquid water is usually strongly attenuated and its backscatter value is hence not representative of the ice cloud properties. A forward model is applied to this state vector to compute simulated values of the radar reflectivity ($Z_{fwd}$) and the lidar apparent backscatter ($\beta_{fwd}$). The state vector is updated until convergence is achieved (when $Z_{fwd}$ and $\beta_{fwd}$ are close enough to $Z$ and $\beta_a$ observations or when iterations do not produce better results).

The state vector is composed of visible extinction ($\alpha_v$) [m$^{-1}$.sr$^{-1}$], lidar extinction-to-backscatter ratio ($S$) [sr] and $N'_0$. $N'_0$ is a variable related to the normalized number concentration parameter $N_0^*$ [m$^{-4}$] via the relationship $N'_0 = N_0^*/\alpha_v^b$, with the



b coefficient determined from in-situ microphysical measurements (see DH0810). The choice of the state vector lies in the fact that first, in the geometric optics limit, $\alpha_v$ has the advantage to be directly linked to the lidar measurements. Then, the lidar apparent backscatter at range $r$ from the instrument can be expressed in the single-scattering limit as a function of $\alpha_v$ and $S$:

$$\beta_a(r) = \frac{\alpha_v(r)}{S(r)} \exp^{-2\tau}, \tag{1}$$

where $\tau$ is the total optical thickness of the atmospheric layer between the lidar and range $r$.

Finally, introducing the concept of normalized particle size distribution (PSD), it is possible to find a robust relationship linking $\alpha_v/N_0^*$ to any variable describing the cloud (Delanoë et al., 2005). Indeed, the retrieved cloud properties are all linked to the PSD which represents the concentration of particles as a function of diameter, $N(D)$. Both diameter and concentration can be scaled and it is possible to find a functional form $F$ fitting any measured size distribution appropriately normalized:

$$\frac{N(D_{eq})}{N_0^*} = F\left(\frac{D_{eq}}{D_m}\right), \tag{2}$$

with $D_{eq}$ the equivalent diameter (in meters) of the melted particle computed from the mass of the particle and the density of ice. This equivalent diameter is scaled by the mean volume weighted diameter, defined as the ratio of the $4^{\text{th}}$ and the $3^{\text{th}}$ moments of the PSD, in terms of $D_{eq}$:

$$D_m = \frac{\int_0^\infty N(D_{eq}) D_{eq}^4 dD_{eq}}{\int_0^\infty N(D_{eq}) D_{eq}^3 dD_{eq}} \tag{3}$$

and the number concentration is scaled by $N_0^*$ which can be written as follows:

$$N_0^* = \frac{4^4}{6} \frac{\left(\int_0^\infty N(D_{eq}) D_{eq}^3 dD_{eq}\right)^5}{\left(\int_0^\infty N(D_{eq}) D_{eq}^4 dD_{eq}\right)^4}. \tag{4}$$

The function in equation 2 can be approximated by a two-parameter modified gamma shape $F_{(\alpha,\beta)}$, the two parameters being determined by a statistic of in-situ measurements (see Delanoë et al. (2014) for the detailed expression of $F$). With this normalized particle size distribution and for a given range of $D_m$, it is then possible to create a one-dimensional look-up table

(LUT) linking all the cloud microphysical variables to the ratio of $\alpha_v$ to $N_0^*$. This LUT is used in the forward model within the iterative process, in particular to retrieve $Z/N_0^*$ from $\alpha_v/N_0^*$. The reflectivity is defined following equation 5:

$$Z = \int N(D)\sigma(D) dD, \tag{5}$$

with the scattering cross-section $\sigma(D)$ obtained by the T-matrix method and (Mishchenko et al., 2004) spheroid approximation. Once the optimized cloud profile has been determined, the LUT is also needed to retrieve additional features of the profile,

such as the $IWC$ and effective radius.

The LUT also needs for $D_{eq}$ to be determined for any ice crystal. To do so, we introduce a relationship giving the mass of a particle as a function of its maximum diameter. This relationship is usually described as a power-law of diameter: $M(D) = \gamma D^\delta$ (Brown and Francis 1995; Mitchell 1996; Lawson and Baker 2006; Heymsfield et al. 2010; Erfani and Mitchell 2016, herafter referred to as EM16). For DARDAR-CLOUD, a combination of (Brown and Francis, 1995) and (Mitchell, 1996) for hexagonal





columns was first used in $V_1$. This relationship will be referred to as "BFM" in the rest of the paper. Other relationships are described in the literature, for specific types of clouds, specific crystal habits or temperature range and are discussed by EM16. To account for the dependency of the relationship between $D$ and $M$ on temperature and particle size, EM16 propose to use a $\delta$ coefficient depending on temperature. However, it is difficult to change $M(D)$ in the retrieval scheme upon the cloud type and

the meteorological conditions without risking to bring discontinuity on the retrievals. As a result, we need to assume a single $M(D)$ relationship which can work for most of the situations.

A priori information about the state vector - derived from a climatology of airborne, ground-based and previous satellite measurements - is used to constrain the inverse problem. This is especially useful when only one measurement is available. Indeed, in most cases, when a cloud profile is measured by both radar and lidar, the vertical fraction of the cloud detected by

both instruments is often preceded in the upper layers by a region only detected by the lidar and followed by a region detected by the radar alone in the lower part. In such regions, the algorithm needs additional information to ensure that the state vector tends towards a physical value. In practice, *a priori* information is only necessary for $S$ and $N_0'$ as the extinction is already well constrained by both the radar and the lidar. In addition, whenever available (that is for semi-transparent isolated cloud layers with optical depths $< 3$), the molecular signal located below the cloud and measured by the lidar is also used as an extra

constraint on the extinction (see DH0810).

DH0810 showed that $N_0'$ had a smooth temperature dependence. For this reason, it was chosen over $N_0^*$ as part of the state vector. We chose here to keep the same formalism as DH0810 expressing the a priori of $\ln(N_0')$ as a linear function of temperature ($T$):

$$\ln(N_0') = \ln(N_0^*/\alpha_v^b) = xT + y, \tag{6}$$

with $T$ in °C. Physically, this describes the idea that as the temperature gets warmer, the aggregation processes tend to increase the size of the particles and reduce their number ($x < 0$).

The lidar ratio (inverse of the value of the normalized phase function at $180\,°$) is a function of many microphysical parameters such as the particle size and shape as well as its orientation (Liou and Yang, 2016). Those variables are expected to vary through the cloud profile. The total attenuated backscatter signal alone, measured by CALIOP, is not enough to give information on this

height dependence. However, to account for the variation of the lidar ratio along the cloud profile, following Platt et al. (2002), $\ln(S)$ is assumed to vary linearly with temperature (in °C):

$$\ln(S) = a_{\ln S}T + b_{\ln S}. \tag{7}$$

This parameterization allows to use the coefficients $a_{\ln S}$ and $b_{\ln S}$ to represent the lidar ratio in the state vector and simplify the iteration process. An *a priori* value is determined for each of those two coefficients.

The errors ascribed to the *a priori* represent how strong this constraint is: the larger the error on the *a priori*, relative to the measurement error, the less relevant the difference between the actual value of the state vector and the *a priori* is and the more the state vector will be allowed to move away from it. The straightforward way to account for the uncertainty on the *a priori* information is to use an error covariance matrix with constant diagonal terms, assuming the confidence we have in





this information is the same everywhere in the cloud profile. When both instruments are available, hopefully the confidence in the measurements is higher than in the *a priori* and the algorithm does not rely on this information. On the contrary, in regions where only one instrument is available, the retrieved values of $S$ and $N_0'$ would essentially be determined by the *a priori*. Therefore, to allow the information from synergistic regions to propagate towards regions where fewer measurements

are available, additional off-diagonal elements are added to the error covariance matrix of the $N_0'$ *a priori*. Those off-diagonal terms decrease exponentially as a function of the distance and aim at describing a spatial correlation in the difference between the actual value of $N_0'$ and its *a priori*. This spatial correlation in the retrieval of $N_0'$ is of course transmitted to the other cloud variables through optimal estimation. More details can be found in Delanoë and Hogan (2008).

The method described above has remained unchanged since the creation of the DARDAR-CLOUD products. In this paper,

we only show improvements that were made and test different microphysical parameterizations in the forward model.

## 3 New parameterizations

This article presents the upgrade of the DARDAR-CLOUD product after the DARDAR-MASK product was modified (Ceccaldi et al., 2013). In this section we describe the improvements on the lidar ratio *a priori* and the microphysical model used in the retrieval method, before quantifying their impacts in the next section.

### 3.1 A priori information for the lidar ratio

In DARDAR-CLOUD v2.1.1 the *a priori* relationship linking $S$ to the temperature was $\ln(S) = -0.0237T + 2.7765$, with $T$ in °C. This was found to produce values of $S$ that are too large at cold temperatures (up to 120 sr) whereas according to the climatology (Platt et al., 1987; Chen et al., 2002; Yorks et al., 2011; Garnier et al., 2015) it should rarely exceed 50 sr. In order to rectify this problem and produce more sensible retrievals, a new *a priori* relationship was determined for $S$. To do so, a linear

regression is performed on the distribution of the retrieved $\ln S$ as a function of temperature, using only lidar-radar synergistic areas. In such regions, the retrieval of $S$ is expected to be well constrained by the measurements. To be even less dependent on the a priori, the old parameterization is kept, but with an error on the slope coefficient ($a_{\ln S}$), multiplied by a factor 10. To produce the statistic of lidar ratio used in this study, the Varcloud algorithm was run on 10 days of CloudSat-CALIPSO observations of the year 2008. The results of the regression are presented in Figure 1. The regression was performed on the

logarithm of $S$. The large majority of points is located in regions where the temperature ranges from $-55\,°C$ to $-20\,°C$, which are the temperatures for which synergistic measurements are statistically most likely to be found. In this domain of temperatures, one can see that the mean and median values of lidar ratio for the different temperature bins are almost identical and fairly close to the first mode of the distributions, which allows for a good assessment of the lidar ratio, as shown by the result of the linear fit. On the contrary, except for the warmest temperatures (above $-30\,°C$), the old parameterization clearly

overestimates the lidar ratio. For colder and warmer temperatures (below $-55\,°C$ and above $-20\,°C$, respectively) the slope of the mean curve changes, with lidar ratio shifting to values $<30$ sr. This leads to a rather low correlation coefficient (-0.3) for the linear regression. Indeed, the fitting process is mainly constrained by the central region where most of the data is found, and



therefore cannot account for the different behaviour of the lidar ratio at the edges of the temperature domain. This illustrates the fact that the variation of the lidar ratio along the cloud profile cannot only be described by the temperature. The comparison of this study to the one from Garnier et al. (2015) confirms this: they are in good agreement where the temperature domains overlap. But as only cold semi-transparent cirrus measured by the lidar and the radiometer are represented in Garnier et al.

(2015), the behaviour is different and the lidar ratios retrieved at temperatures below $-60\,^\circ$C are lower (up to 50% lower at $-70\,^\circ$C).

However, this approximation appears to be legitimate in the lidar-radar areas and is considered valid as *a priori* information on the entire profile, even though larger errors can be expected in lidar-only regions. The final coefficients are chosen to be: $a_{\ln S} = -0.0086$ and $b_{\ln S} = 3.18$, as reported in Figure 1. The reduction of the slope coefficient should prevent the occurrence

of too high values for S at the coldest temperatures.

## 3.2 The microphysical model

The microphysical model is based on 3 main parameterizations: the normalized PSD, the *a priori* of $N_0'$ and the mass-diameter relationship.

As previously said, for DARDAR-CLOUD $V_1$, the mass-diameter relationship used ("BFM") is a combination of (Brown

and Francis, 1995) and (Mitchell, 1996) for hexagonal columns. The Brown and Francis relationship was validated on direct measurements of $IWC$, using a Total Water Content probe combined with a Fluoresence Water Vapor Sensor. However, those measurements were restricted to a couple of flights performed in April, 1992 over the North Sea and to the southwest of the UK, providing a dataset of less than 3000 points recorded at temperatures between $-30\,^\circ$C and $-20\,^\circ$C. The parameterizations of the PSD and the *a priori* were determined using the in-situ dataset described by Delanoë et al. (2005). The main caveat of

this study is that it did not use direct measurements of $IWC$, which may question the reliability of the validation of the microphysical model.

Delanoë et al. (2014) presents a large in-situ dataset collected during several ground-based and airborne campaigns between 2000 and 2007. During those campaigns, direct measurements of $IWC$ were performed with a Counterflow Virtual Impactor or a Cloud Spectrometer and Impactor (CVI/CSI). Such instruments provide valid measurements in the range from $0.01\,\mathrm{g\,m^{-3}}$

to $2\,\mathrm{g\,m^{-3}}$. For a better quality control of the measured PSD, the shattering effect was also considered in this study. Parallel to this, a series of $M(D)$ relationships, based on power-laws of diameter, has been derived by Heymsfield et al. (2010) for specific cloud conditions. Delanoë et al. (2014) compared the measured $IWC$ to the $IWC$ retrieved using the measured PSD and one of those power laws, which allowed to select for each campaign the $M(D)$ relationship giving the best match to the measured $IWC$. The selected $M(D)$ are presented in (Delanoë et al., 2014, Table 3). The general mass-size parameterization,

specific to this dataset and made of different power-laws as a function of the measurement campaigns, will be referred to as the "BEST" parameterization.

In the following, we present how this dataset was used to refine the microphysical model of the Varcloud algorithm.




### 3.2.1 The normalized PSD

The normalized particle size distribution is updated with the new coefficients determined by Delanoë et al. (2014) using a least square regression on two moments of the PSD, namely the visible extinction, $\alpha_v$, and the radar reflectivity, Z. To do so, a mass-size relationship had to be assumed and the "BEST" parameterization was chosen. Figure 2 compares the shape of the normalized PSD obtained with the old parameterization ($\alpha$=-2, $\beta$=4) to the new update ($\alpha$=-0.262, $\beta$=1.754). Those new coefficients mainly impact the very small diameters and the tail of the distribution. The center of the distribution (around $D_{eq}/D_m = 1$) remains almost unchanged. However, the new normalized PSD is now characterized by higher values of normalized number concentration for the largest particles. This could increase the impact of the change in the mass-diameter relationship. Additionally, it is reminded here that in a first-order approximation, the radar reflectivity is more sensitive to the size of the particles whereas the lidar backscatter depends mainly on the concentration. As a result, if the weight on the large particles is increased, a higher sensitivity can be expected in regions detected by the radar.

### 3.2.2 The a priori of $N_0'$

As mentioned previously, the *a priori* value for $N_0'$ is obtained via a parameterization as a function of the temperature. Following equation 5, three parameters $(x, y, b)$ have to be determined. The first two are used to link $N_0'$ to the temperature and the last one, $b$, relates $N_0'$ to the normalized number concentration $N_0^*$. To do so, several linear regressions are performed between $N_0'$ and $T$ to identify the $x$ and $y$ parameters for different values of $b$. The values of $N_0^*$ and $\alpha_v$ are retrieved from the measured PSDs and the "BEST" $M(D)$. The final set of parameters $(x, y, b)$ is chosen with the highest coefficient of determination $R^2$. For this study, the "Subvisible" class in the dataset presented in Delanoë et al. (2014) has been removed as it consists in very small crystals associated to very cold temperatures and we considered that it was too far from the main common radar-lidar domain in terms of temperature conditions. The data points measured at temperatures above $-15\,°C$ during the MPACE campaign have also been removed.

Figure 3 shows the result of the regression. The new values for the parameters $(x, y, b)$ are presented in panel a). The new *a priori* parameterization for $N_0'$ as a function of $T$ is very close to the old version (panel b). The main difference is for the $b$ coefficient which leads to an increase in the *a priori* of $N_0^*$ of almost 2 orders of magnitude.

### 3.2.3 The mass-diameter relationship

For each specific campaign, the optimal coefficients of a power-law could be determined to describe the mass-diameter relationship. This was done by comparing the bulk $IWC$ to the retrieved $IWC$ obtained by the combination of the measured PSD and this power-law (Heymsfield et al., 2010), leading to the creation of the "BEST" $M(D)$ parameterization. Figure 4 shows the comparison between the measured $IWC$ and the retrieved $IWC$, for different mass-diameter relationships: the "BEST" parameterization, where a specific power-law is selected for each campaign (left), the parameterization used in DARDAR-CLOUD $V_1$, namely "BFM", applied to all the campaigns (middle), and finally the "Composite" parameterization, also applied to the entire dataset (right). The "Composite" was developed by Heymsfield et al. (2010) using the measurements of all cam-





paigns, combining different types of clouds and situations. As we want to keep a single $M(D)$ in our algorithm, it is interesting to compare this more recent parameterization to "BFM".

It is clear that using dedicated parameterizations for specific atmospheric conditions and/or cloud types (that is the "BEST"
parameterization) gives better results when comparing the model to the measurements. However, in the framework of our retrieval scheme, we prefer to use one parameterization which gives the best fit, on average. Hence the choice of "BFM" for DARDAR-CLOUD $V_1$. As presented in Figure 4, panel b, this parameterization critically underestimates the measured $IWC$, especially for values above $0.1\,\mathrm{g\,m^{-3}}$. With the "Composite" relationship on the contrary, it is possible to improve the match with the measured $IWC$ (panel c). It was therefore decided to modify Varcloud's microphysical model and use "Composite"
instead of "BFM". Details of these two relationships can be found in Table 1. The main difference between the expressions of "BFM" and "Composite" is the power coefficient: for particles $> 100\,\mu\mathrm{m}$, this coefficient equals 1.9 for "BFM" and 2.2 for "Composite". As a result, for a given mass, the "Composite" relationship provides a smaller equivalent diameter for the ice crystal than "BFM". This difference increases when the mass and the size get larger. On the contrary, for small diameters ($\leqslant 100\,\mu\mathrm{m}$), "BFM" creates denser particles with smaller $D_{eq}$. Referring to EM16, these $\delta$ coefficients are in the domain of
optimal values for ice crystals from continental ice clouds, at temperatures between $-60\,^{\circ}\mathrm{C}$ and $-20\,^{\circ}\mathrm{C}$ and of size ranging from $100\,\mu\mathrm{m}$ to $1000\,\mu\mathrm{m}$. Moreover, EM16 showed that the "Composite" $M(D)$ conformed closely to their fit performed on measurements from the SPARTICUS campaign.

## 4   Evolution in DARDAR-CLOUD retrievals: comparison between $V_1$ and $V_2$

DARDAR-CLOUD $V_1$ was created using the DARDAR-MASK v1 classification to select the hydrometeors on which to per-
form the retrieval. Since the classification was updated with Ceccaldi et al. (2013) (DARDAR-MASK v2), we will briefly show in a first instance the impact of the change in classification on the microphysical properties.

Considering new information obtained from larger and more recent in-situ datasets, several parameters were changed in the forward model in order to improve the results in a global scale. We will, in a second instance, show how those modifications impact the $IWC$.

The analysis is made over the same 10 days (~3M profiles) of CloudSat-CALIPSO observations as those used to determine the new *a priori* for the lidar ratio. The detail of this dataset is presented in Table 2. All the studies presented in this paper were performed using the same set of observations.

### 4.1   Impact of the new classification

As detailed in Ceccaldi et al. (2013) the new hydrometeor classification (DARDAR-MASK v2) reports fewer ice clouds in the
upper troposphere than DARDAR-MASK v1. This is due to the fact that the new methodology is more restrictive in creating the lidar mask in order to include as few noisy pixels as possible. On the other hand, it can miss some very thin ice clouds. Also, the false cloud tops detected by the radar due to its original resolution have been removed from the radar mask; hence fewer fake ice pixels are retrieved on radar-only data on top of lidar-radar pixels.



To study the impact of the new classification on the retrieved $IWC$ we run the algorithm with the DARDAR-CLOUD $V_1$ configuration with both the old and the new classifications. The distribution of derived $\log_{10}(IWC)$ as a function of temperature is then compared.

The distributions are computed as the histogram of occurrence (as percentage of pixels included in the retrieval) of $\log_{10}(IWC)$ in temperature bins of $0.5\,°\mathrm{C}$ in the range $-88\,°\mathrm{C}$ to $0\,°\mathrm{C}$. The comparison between the two distributions is displayed in Figure 5. We can see that using the new classification globally leads to fewer pixels included in the retrieval, especially for $IWC$ lower than $10^{-2}\,\mathrm{g\,m}^{-3}$ (Figure 5, a). Consequently the mean $\log_{10}(IWC)$ decreases more rapidly with decreasing temperature than when using the old classification (Figure 5, b). This observation is consistent with the fact that the new classification is more restrictive; lidar noisy pixels and very thin ice clouds pixels producing very low $IWC$ are not included in this distribution any longer, leading to higher mean values. This is highlighted by the comparison of both distributions in the lidar-only region (Figure 5, d). It is very clear that fewer pixels are selected in the new version, especially for $IWC < 10^{-2}\,\mathrm{g\,m}^{-3}$. There is also fewer pixels of low $IWC$ in the radar-only regions (panel e) due to the suppression of fake cloud top detection on the radar signal.

This new selection of cloud pixels on the lidar signal also affects the synergistic areas. Indeed, if fewer lidar pixels are detected, then the number of lidar-radar pixels decreases in favor of radar-only pixels. In such regions, most of the pixels that were removed from the lidar cloud mask are suspected to be noisy measurements. Including noise into a variational retrieval can increase its instability and lead to higher errors. It is therefore safer to have fewer but more reliable pixels in common for the two instruments. On the other side, the number of higher $IWC$ values ($> 10^{-2}\,\mathrm{g\,m}^{-3}$) is slightly enhanced. The way the new categorization better deals with the radar's ground clutter could account for more radar and lidar-radar areas detected as ice clouds close to the ground, with temperatures between $-10\,°\mathrm{C}$ and $0\,°\mathrm{C}$.

When comparing the two configurations pixel by pixel, one can see that no bias is introduced by the new classification as the histogram of differences is centered on 0%. As a consequence, the increase in the mean of retrieved $IWC$ is solely due to the removal of pixels of very low $IWC$ values. The 18% of the data showing -100% difference account for pixels that used to be classified as ice in the old configuration and that are not detected by the new algorithm because they are suspected of being noisy pixels. Pixels that are not affected by the new classification show in average the same values of $IWC$. Overall, more than 50% of the data show less than 20% difference. Larger differences appear in profiles where ice pixels were removed or added, which potentially changed the balance between the instruments.

For the following studies, the algorithm is applied to the new classification and both instruments are used whenever available.

## 4.2 Impact of the new a priori relationship for the lidar ratio

To be more consistent with the extinction-to-backscatter ratio ($S$) values found in the literature and to account for the assessment made on the retrieved $IWC$ in high troposphere (Deng et al., 2013), a new *a priori* was determined using the well-constrained retrievals from the radar - lidar synergistic areas (section 3.1). To assess the impact of this new configuration on the retrievals, the Varcloud algorithm was run using one after the other the two different *a priori* relationships for $S$.



### 4.2.1  $S$ **retrievals**

With the new parameterization, one can see that the values are, in average, smaller, and more centered around an average value
of $35\,\mathrm{sr}$ (Figure 6a). As a result, contrary to what could be found with the old configuration, the maximum values does not

exceed $60\,\mathrm{sr}$. Panels b) to j) show the distributions (in % of the total number of retrieved pixels) of $S$ for the two different
configurations as well as the distributions of their relative difference $\left(\frac{S_2 - S_1}{S_1}\right)$ as a function of temperature. As a consequence
of this new configuration, the retrieved lidar ratio tends to be closer to the *a priori* value. This new parameterization was
determined using former Varcloud retrievals, therefore it is logical that the fit to the algorithm is better. This is particularly
visible when comparing panels c) and f) for lidar-only areas.

As the algorithm only returns the two coefficients of the relation linking $\ln S$ to $T$, the retrieved lidar ratio depends on the
measured profile as a whole. For stability reasons, a small error was set on the *a priori* in the cost function. As a result, the
lidar ratio mainly follows the *a priori* information. But it is allowed to move away from it, especially in synergistic areas
and when lidar pixels can receive information from lidar-radar areas or from the molecular signal located below optically thin
clouds. This explains the two modes observed on the relative differences distributions. One mode is closer to 0% difference

(between -25 and +25%) and corresponds to profiles where the lidar ratio is determined from the synergy of radar and lidar or
constrained by molecular signal and less constrained by the *a priori* information. As a consequence, the new parameterization
has less impact in these regions and the dispersion of the signal accounts for the amount of additional constrain available on
each profile provided by the number of radar pixels collocated with the lidar or the possibility to use molecular signal. The
second mode follows a thin line representing the difference between the two *a priori* slopes and contains profiles where the *a*

*priori* has the major influence in determining the lidar ratio e.g. profiles with lidar measurements alone. This could be further
reduced using additional sensors such as IR or/and visible radiometers.

### 4.2.2  $IWC$ **retrievals**

These differences in lidar ratio can impact the ice water content via the visible extinction. Differences in $\log_{10}(IWC)$ distribution are shown in Figure 7. As expected, changing the configuration only impacts $IWC$ below $10^{-1}\,\mathrm{g\,m^{-3}}$. Indeed, we

expect $IWC$ above this threshold to be found in the lower parts of the clouds, where only the radar can provide measurements
and therefore the impact of the lidar ratio *a priori* can be neglected. The global distribution of $\log_{10}(IWC)$ is shifted towards
lower values, and the lower the $IWC$, the more differences can be seen.

A more detailed comparison is made in Figure 8. It is clear that $IWC$ tends to increase with temperature as well as its
variability (Figure 8, a). For lidar-only pixels, information is mainly available at temperatures below $-40\,^{\circ}\mathrm{C}$ (Figure 8, b). In

most cases, the lidar is strongly attenuated when it penetrates deeper in the cloud to reach higher temperatures. Low level ice
clouds can be detected by the lidar but only if the attenuation is not too strong in the higher levels, which is the case for only
a minority of the cloud scenes detected by the CloudSat-CALIPSO instruments. In cold regions detected by the lidar alone,
$IWC$ values range from $5 \times 10^{-4}\,\mathrm{g\,m^{-3}}$ for temperatures below $-80\,^{\circ}\mathrm{C}$ to almost $10^{-1}\,\mathrm{g\,m^{-3}}$ around $-60\,^{\circ}\mathrm{C}$. Radar-only
pixels can be found for temperatures above $-50\,^{\circ}\mathrm{C}$ where $IWC$ from $10^{-3}\,\mathrm{g\,m^{-3}}$ to $1\,\mathrm{g\,m^{-3}}$ can be observed, especially in





the warmest regions where $T > -20\,^{\circ}\text{C}$ (Figure 8 c). Finally, synergetic areas are found in between those two regions (Figure 8 d). When looking at the difference between $V_1$ and $V_2$ (Figure 8 e-h), red areas indicate that more pixels from $V_2$ were found to fit in the corresponding $[IWC - T]$ range than from $V_1$. On the contrary, in blue areas, there are fewer pixels from $V_2$. One

can see again that the distribution is shifted towards lower values of $IWC$ no matter where in the cloud and which instrument is available. However, the difference is the strongest at the coldest temperatures $(<-40\,^{\circ}\text{C})$ which is where we find most of the lidar-only pixels and where the difference between the two lidar ratio *a priori* relationships is the largest. At warmer temperatures, on the contrary, there is almost no change in the $\log_{10}(IWC)$ distribution as the retrieval mainly depends on the radar measurements. Following the behaviour of the lidar ratio, 2 modes can be distinguished in the distribution of relative

difference in $IWC$ as a function of temperature (Figure 8 i-l). Most of the $IWC$ retrievals present differences less than 25%. However, for temperatures between $-50\,^{\circ}\text{C}$ and $-70\,^{\circ}\text{C}$, where most of the lidar-only pixels can be found, the discrepancies vary between -40% and -50% in average.

### 4.3 Impact of the new microphysical model

The analysis of a more recent and larger in-situ dataset including bulk $IWC$ measurements allowed the microphysical model

to be refined as explained in section 3.2. In this section, we show the consequences of this new parameterization in the $IWC$ retrievals. To do so, the Varcloud algorithm was run using one after the other $V_1$ and $V_2$ LUT and $N_0'$ *a priori*, both associated with the $V_2$ lidar ratio *a priori*. In the same way as for the study on the new lidar ratio *a priori*, we can look at the differences in the distribution of $\log_{10}(IWC)$ (Figure 9). The impact of the microphysical model is more complex as its action occurs both in the radar forward model and at the end of the process when the $IWC$ is retrieved from extinction and $N_0^*$ thanks to

the 1D-Lookup table. Moreover, the interactions that may exist between the parameters that were refined (the PSD, $M(D)$ and $N_0'$ *a priori*) are likely to have different impacts on the retrieval depending on the physical and microphysical conditions of the observed cloud region. As a result, we will not try to interpret here the differences observed between the two microphysical configurations but describe how the retrieval is impacted.

First of all, when looking at Figure 9, panel e), it seems that the impact of the new microphysics strongly depends on the

temperature, with an increase in the averaged retrieved $IWC$ for temperatures below $-40\,^{\circ}\text{C}$ and a decrease for temperatures above $-40\,^{\circ}\text{C}$. When pixels are separated in different regions depending on the available instruments (lidar only, radar only or both), it is clear that the impact of the new model is also very different for the two instruments: the increase of $IWC$ observed for the cold temperatures is associated with lidar-only pixels (panel f). On the contrary, radar-only pixels are marked by a shift of the distribution towards lower values of $IWC$. Where both instruments are available, the opposite effects cancel each other

out, which leads to almost no difference in the distribution of $\log_{10}(IWC)$ in such regions (panel h). The differences observed in the retrieved $IWC$ in regions detected by the two instruments barely exceed 10% (panel l). On the contrary, in regions where only one instrument is available, differences are observed between 0% and 40% for lidar-only pixels and between -40% and 0% for radar-only pixels.

For pixels detected by the lidar only, two modes can be observed in the distribution of the differences between $V_1$ and $V_2$, which overall leads to a decrease in the retrieved ice water path. The main mode is the thin red (strong occurrence) curved line



and accounts for profiles where only the lidar was able to detect a cloud. In such conditions, the extinction is retrieved using the lidar measurement and the lidar ratio *a priori*. It is therefore completely independent on the microphysical model. The normalized concentration number parameter $N_0^*$ is then derived using the extinction and the *a priori* of $N_0'$. As a result, the

retrieved $IWC$, derived using the extinction and the LUT, depends on the microphysical model in a deterministic way. This curve is the direct translation of the difference between the 2 configurations into the relationship between visible extinction and $IWC$ as it is parameterized in the LUT. It also illustrates the strong dependency of the microphysical parameterization on the temperature. The second mode presents smaller differences and accounts for the influence of radar measurements deeper in the cloud profile, which balance the increase in $IWC$ by their opposite effect. For radar-only pixels, the influence of the

microphysical parameterization is more diffuse as it also plays a role in the iteration process through the radar forward model.

### 4.4   Conclusions regarding DARDAR-CLOUD new version

#### 4.4.1   $IWC$ retrievals

As a summary of all these modifications in the retrieval code, Figure 10 presents the new distribution ($V_2$) of retrieved $\log_{10}(IWC)$ (panels a to d) and its difference with the distribution of DARDAR-CLOUD $V_1$ (panels e to h) as well as the

relative differences in $IWC$ between the two versions (panels i to l). The new version includes all the updates presented above. Also taken into account is the update of CALIPSO Level 1 products (v4) consisting of the use of better ancillary data sets: a more accurate DEM (Digital Elevation Model) and a new reanalysis product for the atmospheric variables (MERRA-2), which is shown to allow for more reliable CALIOP calibration coefficients. Information on this update can be found on the NASA website at the following address: https://www-calipso.larc.nasa.gov/resources/calipso_users_guide/data_summaries/l1b/CAL_

LID_L1-Standard-V4-10.php

When comparing the two distributions of $\log_{10}(IWC)$, we can see that the reduction in the number of retrieved $IWC$ pixels due to the new classification prevails in lidar (panel f) and lidar-radar areas (panel h). On the contrary, in regions where only radar measurements are available, more pixels are retrieved (panel g). Different features can be observed in the relative differences distributions (i-l), which are the combination of the updates in the microphysical model that strongly modify the

retrievals in the radar-only regions and the impact of the new lidar ratio *a priori*, mainly affecting the lidar-only and lidar-radar areas. In these areas, the influence of the new LUT is opposed to that of the lidar ratio *a priori*: the new normalized PSD associated to the choice of the "Composite" mass-size relationship produces higher values of $IWC$ when lower values of $S$ tend to create lower $IWC$. It appears, however, that the influence of the lidar ratio prevails, visible in the two modes that can be observed in panel j, similar to the ones discribed in section 4.2. The combination of all the modifications made

on the retrieval algorithm also seems to create larger differences, positive as well as negative, regardless of the pixel location. However, the probability of occurence for such values is much lower than for the features previously described. The relative differences shown here are calculated only where ice is detected by both configurations. It is in the synergistic areas that the highest probability is found for the smallest differences.





Figure 11 shows the global histogram of the relative difference in $IWC$ between DARDAR-CLOUD $V_1$ and the new version (a) and the contribution of the different updates. This information was obtained by running the algorithm several times with a different configuration. Each histogram is a comparison between two retrievals, processed with only one modification in the algorithm: changing the version of the CALIPSO Level 1 product (b), the DARDAR-MASK classification product (c), the

parameterization of the lidar ratio *a priori* (d) or the microphysical model (e). When each contribution is taken separately, it can be seen that the highest percentage of occurrence is found for differences <5%. However, the combination of the new *a priori* for the lidar ratio and the new microphysical model leads to an average reduction of -20% from DARDAR-CLOUD $V_1$ to DARDAR-CLOUD $V_2$. As said previously in section 4.1, the 18% of the data showing -100% difference accounts for the evolution of the hydrometeors classification (Figure 11, a and c). The new updates on CALIPSO product can also modify the

classification and the retrieval, although to a lesser extent. Indeed, more than 80% remain with differences <5% (Figure 11, b). The largest differences are due to the impact of the new classification, which accounts for the broadening of the probability density observed in Figure 10. This analysis shows that less than 10% of the data remain with differences <5%.

### 4.4.2    $r_e$ retrievals

Particle size information is given in DARDAR-CLOUD via the retrieved effectif radii ($r_e$). $r_e$ is defined as the ratio of $IWC$

and $\alpha_v$:

$$r_e = \frac{3IWC}{2\alpha_v\rho_i}, \tag{8}$$

with $\rho_i$ the density of solid ice. Figure 12 shows the new distribution ($V_2$) of retrieved $r_e$ (panels a to d) and its difference with the distribution of DARDAR-CLOUD $V_1$ (panels e to h). The relative differences in $r_e$ between the two versions is also presented (panels i to l). Similarly to $IWC$, the effective radius tends to increase with temperature as well as its variability.

The influence of temperature is however stronger as the dispersion of the retrieved $r_e$ is much smaller than that of the retrieved $IWC$. The new parameterizations clearly impact the retrieved $r_e$: the entire distribution as a function of temperature is shifted towards larger values, reaching $140\,\mu m$ in $V_2$ for the warmest regions when in $V_1$, the highest value of retrieved $r_e$ was around $100\,\mu m$ . This effect is due to the change of microphysical model which has the strongest influence on the retrieval of $r_e$. For cold temperatures ($<-60\,°C$), the differences are slightly smaller due to the influence of the lidar ratio *a priori* which

has an opposite effect. The largest differences (between +20% and +40%) are found in the radar-only regions at the warmest temperatures. For pixels that benefit from the combined influence of the two instruments, the impact of the configuration change is reduced (differences are found between +5% and +25%).

### 5    Summary and discussion

This paper gives an overview of the main characteristics of the DARDAR-CLOUD new version, describing the modifications

made on the Varcloud algorithm and their consequences on the retrieved ice water content. We have shown that the evolution of the DARDAR-CLOUD forward model configuration and the DARDAR-MASK hydrometeors classification could lead to



differences in retrieved $IWC$ of up to a factor 2 relative to the earlier release, regardless of the instruments available or the temperature range. These very large discrepancies, which are mainly the consequence of the new phase categorization, represent 5% of the data used for this study. 90% of the $IWC$ values show differences less than 50% with the old configuration. The change in the microphysical model also affects the retrieved $r_e$ everywhere along the temperature profile, with differences

ranging from 5% to 40%.

The new values in the parameterization of the lidar extinction-to-backscatter ratio *a priori* was shown to have little influence on the retrieved $r_e$. On the other hand, for $IWC$ retrievals, they have more impact for temperatures below $-40\,°C$ and induce lower $IWC$ (up to -50% for the coldest temperatures) in every cloud region detected by the lidar. However, their impact is significantly reduced by the new LUT which introduces opposite modifications in lidar-only regions. Radar-only regions are

mainly influenced by the modifications of the LUT and the *a priori* of $N_0^*$, which also reduce the values of $IWC$ up to -40% for the warmest temperatures. In synergistic areas, the combination of the two instruments seems to mitigate the impact of the modifications made in the microphysical model. Nevertheless, differences between -20% and 20% are also found in this region between $-60\,°C$ and $-20\,°C$

Trying to find a simple parameterization of the lidar extinction-to-backscatter ratio was shown to be rather challenging and

uncertainties remain high, particularly in regions where synergies are not available. More work could be done on the subject, adding radiometric instruments or looking at new instrumental platforms, such as the upcoming ESA/JAXA EarthCARE satellite, whith a more sensitive radar and High Spectral Resolution Lidar which could help refine our analyses.

This sensitivity study was done to help us identify improvements to be considered in the new version that will be made available at ICARE/AERIS data centre. Our approach here is to use information and datasets validated by the litterature to de-

termine the microphysical assumptions and study the sensitivity of our algorithm to those assumptions. Further improvements are aimed at, relying on more in situ and satellite observations to make parameterizations and combination of instruments more efficient benefiting from Calipso-CloudSat extension and EarthCare advent.

*Competing interests.*   no competing interests are present

*Acknowledgements.*   Quitterie Cazenave's research is funded by CNES and DLR/VO-R young investigator group and we thank the ICARE Data and Services Center (http://www.icare-lille1.fr) as well as the CloudSat and CALIPSO projects for providing access to the data used in this study.





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





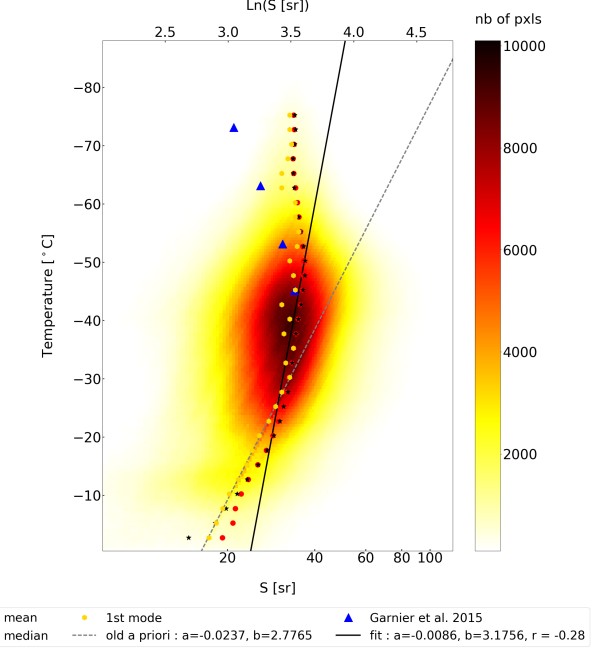

**Figure 1.** Linear regression on the probability density distribution of $\ln(S)$ as a function of temperature. The corresponding values of S in steradian are also displayed. The result of the linear regression fit is represented in solid lines and the old *a priori* relationship in dashed lines. Red and yellow dots are the median and first mode (respectively). The parameterization obtained with the retrievals from Garnier et al. (2015) is also displayed for comparison (blue triangles)

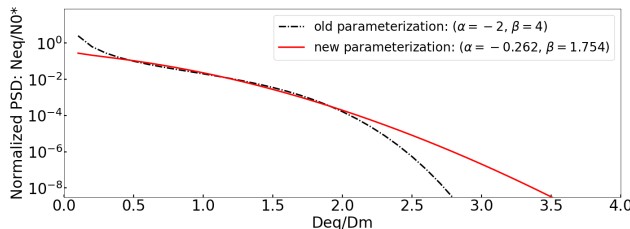

**Figure 2.** Look-up table parameters: the normalized PSD (DARDAR-CLOUD $V_1$ is represented by the black dashed line and the new parameterization, $V_2$, is represented by the red solid line).

**Table 1.** Mass-diameter relationships used for DARDAR-CLOUD $V_1$ and $V_2$.

| | $M(D)$ with M in gramms | | |
|---|---|---|---|
| | $D \leqslant 0.01cm$ | $0.01 < D \leqslant 0.03cm$ | $D > 0.03cm$ |
| BFM (DARDAR $V_1$) | $1.677.10^{-1}D^{2.91}$ | $1.66.10^{-3}D^{1.91}$ | $1.9241.10^{-3}D^{1.9}$ |
| Composite (DARDAR $V_2$) | $7.10^{-3}D^{2.2}$ | | |





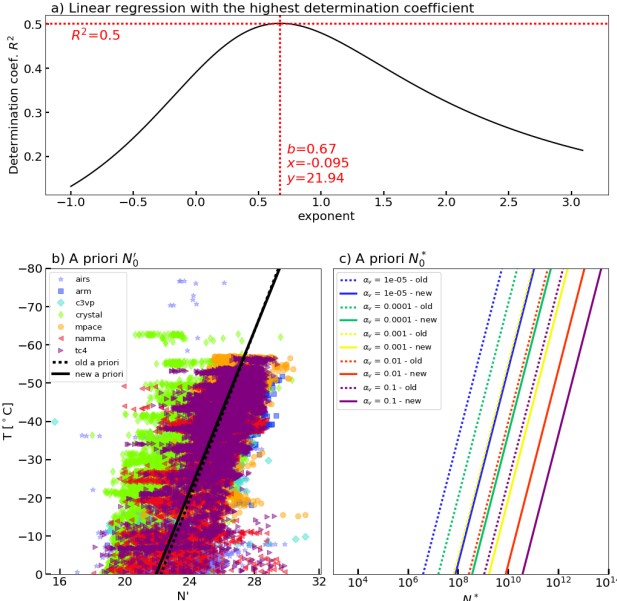

**Figure 3.** Determination of the a priori of $N_0^*$: values of the coefficient of determination $R^2$ for different values of $b$ and values of the coefficients for the best fit (a), result of the linear regression on $N_0'$ for the entire dataset as a function of temperature and comparison with the old parameterization (b) and corresponding $N_0^*$ a priori for different values of extinction (c).

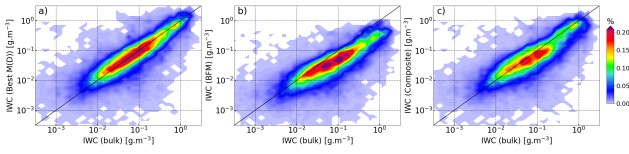

**Figure 4.** Comparison between the measured IWC and the retrieved IWC, using different mass-diameter relationships: best M-D relationship compared to the measured IWC (a), (Brown and Francis, 1995) and (Mitchell, 1996) (b), the "Composite" parameterization (Delanoë et al., 2014) (c)

**Table 2.** CloudSat-CALIPSO observations used in this study.

| Year | Month | Days |
|---|---|---|
| | January | 01/01/2008; 02/01/2008; 03/01/2008 |
| 2008 | February | 01/02/2008; 02/02/2008 |
| | June | 01/06/2008; 02/06/2008 |
| | July | 01/07/2008; 02/07/2008; 03/07/2008 |





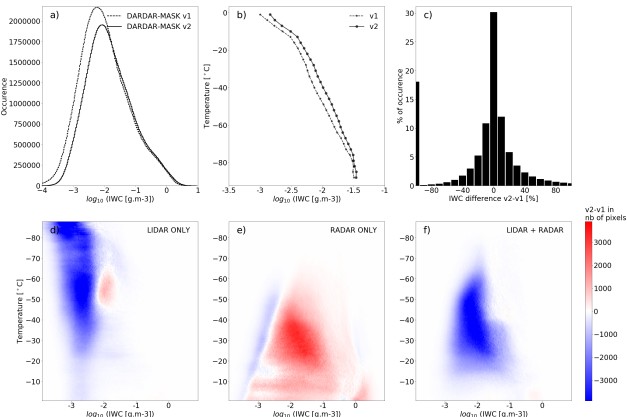

**Figure 5.** Comparison of DARDAR-MASK v1 and DARDAR-MASK v2 in terms of IWC retrieved using Varcloud ($V_1$): histograms of $\log_{10}(IWC)$ (a) with the old version of DARDAR-MASK in dashed line and the new version in solid line, the mean of the distribution of $\log_{10}(IWC)$ as a function of temperature (b) for DARDAR-MASK v1.1.4 (dashed line) and DARDAR-MASK v2.1 (solid line), the histogram of the relative difference $\frac{IWC_{v2}-IWC_{v1}}{IWC_{v1}}$ between DARDAR-MASK v2 and DARDAR-MASK v1 (c) and the difference between the two configurations in terms of number of pixels in each $[T - \log_{10}(IWC)]$ bin for lidar-only pixels (d), radar-only pixels (e) and pixels combining the two instruments (f).

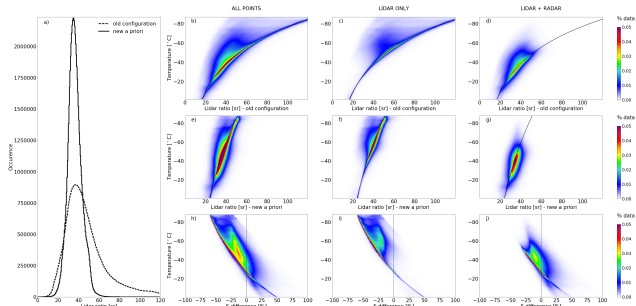

**Figure 6.** Comparison of the retrieved lidar ratios for 2 *a priori* parameterizations: histograms of $S$ (a) with the old *a priori* relationship (dashed) and the new relationship (solid), probability density distributions of $S$ as a function of temperature obtained when using the old configuration (b-d) and the same results with the new configuration (e-g), probability density distribution of the relative difference between the two lidar ratios obtained from the two different configurations at each retrieved pixel (h-j). The black line represents the *a priori*.




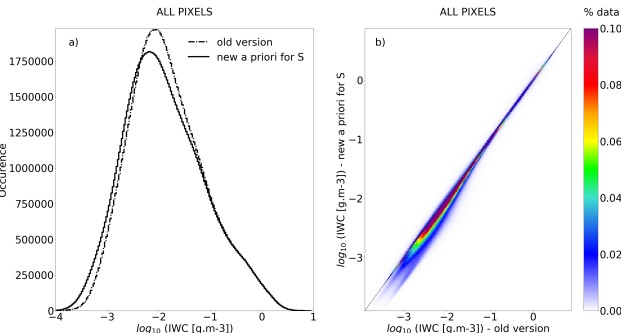

**Figure 7.** Comparison of the retrieved IWC for 2 *a priori* parameterizations of the lidar ratio: panel a) shows the histograms of $\log_{10}(IWC)$ using the old ($V_1$) *a priori* relationship (dashed) and the new ($V_2$) relationship (solid). Panel b) shows the probability density distribution of each IWC retrieval relative to the other.

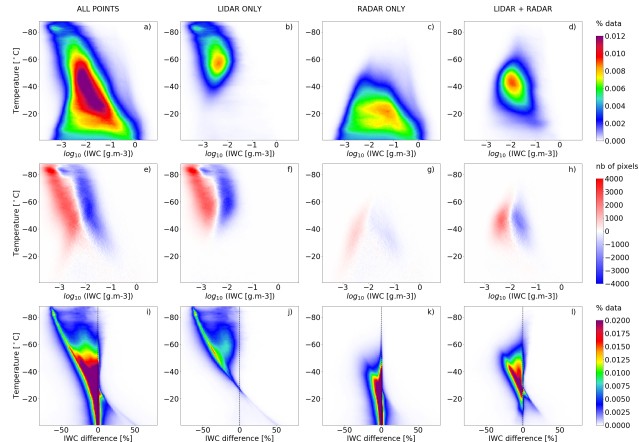

**Figure 8.** Comparison of the retrieved IWC for the $V_1$ and $V_2$ *a priori* parameterizations of the lidar ratio: probability density distributions of $\log_{10}(IWC)$ as a function of temperature obtained when using the new configuration ($V_2$)(a-d), differences in the $\log_{10}(IWC)$ distributions in terms of number of retrieved pixels for each [$\log_{10}(IWC)$-T] bin (e-h), and finally the distribution of the relative difference between IWC obtained with the new $V_2$ lidar ratio *a priori* and IWC obtained with the old $V_1$ configuration as a function of temperature (i-l). First column shows the distributions for all retrieved pixels, second column, for lidar-only pixels, third column, for radar-only pixels and last columns for lidar-radar pixels.



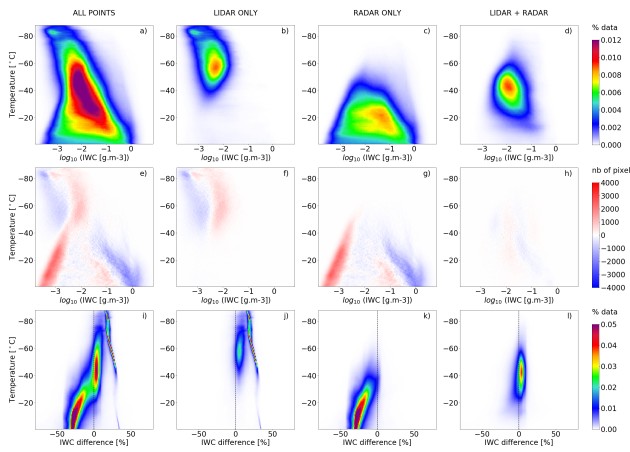

**Figure 9.** Comparison of the retrieved IWC for the 2 microphysical parameterizations presented in 33.2 (but with the same lidar ratio *a priori*): same panels as for Figure 8.

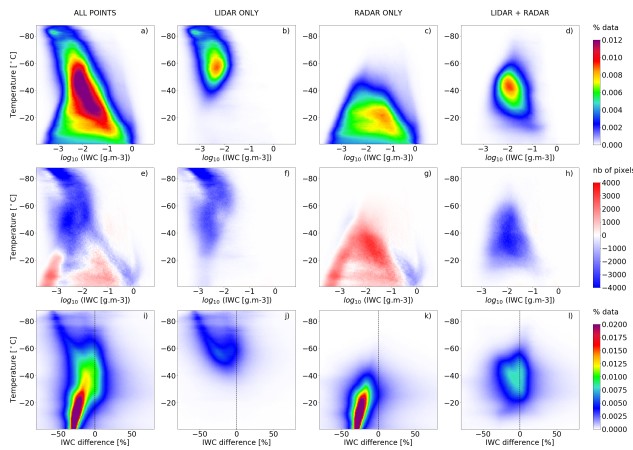

**Figure 10.** Comparison of the retrieved IWC between $V_1$ and $V_2$: same panels as for Figures 8 and 9.





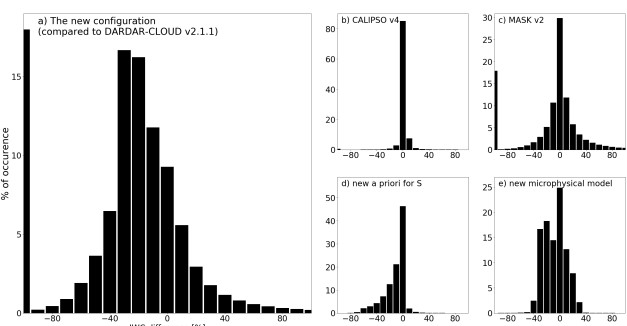

**Figure 11.** Histograms of the relative differences in IWC between $V_1$ and $V_2$ (a) and for every modification made in the new version: CALIPSO v4 (b), DARDAR-MASK v2 (c), the new *a priori* for $S$ (d) and the new Look-up table (e).

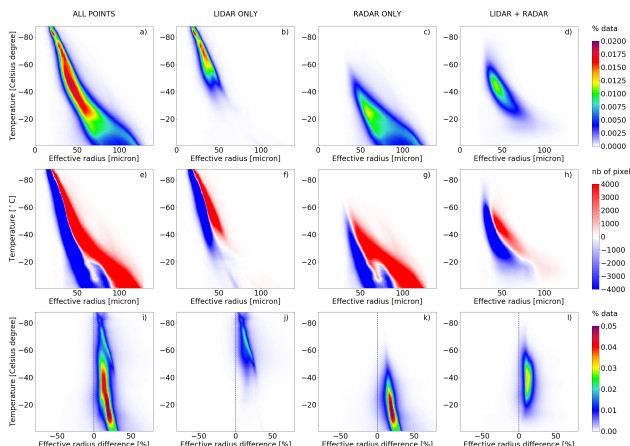

**Figure 12.** Comparison of the retrieved effective radius ($r_e$) between $V_1$ and $V_2$: same panels as for Figures 8, 9 and 10.