# Peer review of "Evolution of DARDAR-CLOUD ice cloud retrievals: new parameters and impacts on the retrieved microphysical properties"

_Atmospheric Measurement Techniques, 2018_

## Referee Comment (RC1) · Anonymous Referee #1 · 16 Jan 2019

Considering the DARDAR product is widely used for modeling community, documentation of the algorithm evolution is critical. Thus, publishing the paper is important. Although there are few issues could be better addressed, the paper is clearly presented. I suggest it for publication after the following issues are properly addressed.

Major issues:

1. In the abstract, the impacts of changes are presented in an inconsistent way. The impact for IWC is provided with the upper range while the range for size is given. Up

to 50% for IWC is a huge difference. It will be better that the impacts are accessed in term of mean changes and ranges.

2. The quality of the figures is poor. The x-axis and y-axis titles are too small, and units also need to be correctly displayed.

3. It is not clear why short periods of data were selected for comparison. It will be great that results on seasonal and global scales are presented to document the impacts due to algorithm changes.

4. The algorithm outputs lidar ratio, which has limited information from observations directly. The V1 lidar ratio a prior was clearly wrong. The V2 results are more reasonable. However, the results of lidar ratio should be compared with CALIPSO results. Also, it is important to discuss on how multiple scattering is treated in the algorithm because it is directly tied to effective lidar ratio selection.

Minor issues:

1. Page 3, line 23: Change "restate" to "summarize".

2. Page 4, Line 3: change "apparent" to "attenuated".

3. Page 4, lines 11-14: the equivalent diameter discussion here is confusion.

4. Page 4, line 24: "the LUT" refers which LUT. Is there only one LUT for the algorithm?

5. Page 5, lines 4-5: The logic does not make sense. Maintaining continuity does not mean accurate results.

6. Page 13, lines 9-10: The two extremes of radar-only results seems indicating that the algorithm for the radar-only region is not very stable.

7. Table 1: use the formal way to represent the constant coefficients.

8. Fig. 5a and 5b: It is hard to separate the two line styles. How about using color lines for them?

9. Fig. 9 figure caption: What does "33.2" mean here?

---

## Referee Comment (RC2) · Anonymous Referee #2 · 19 Feb 2019

This paper presents results of microphysical ice properties from the radar and lidar on board the A-train in view of the latest refinements to the DARDAR cloud product, namely: 1. apriori information on the lidar ratio; 2. an improved particle size distribution for the ice particles; 3. An improved N'o and 4. A new mass-diameter relationship. The A-train data set currently provide our best estimate of the global climatology of ice clouds, so improving this data set is clearly a valuable step forward, and so the results are worthy of publication, subject to clarification of some issues.

Major Points

1. The abstract should be more informative and provide a more precise summary of the findings. At present the statements are too vague. Currently it says IWC can be 'up to 50% with, globally, a reduction'. 50% is a large change. What is the global average reduction? Effective radius increases between 5% and 40%, with the largest difference in clouds between -20C and 0C. The new lidar ratio of 35 +/10 sr for cold clouds is quite a reduction on the previous values. Line one of the introduction stresses the importance of ice clouds on the radiation budget, but this aspect does not seem to be directly addressed in the rest of the paper. Do changes in effective radius for the warmer ice clouds lead to changes in the radiation budget? Perhaps not, as such clouds are already optically thick? Do changes in the lidar ration affect the radiative properties of the thin cold ice clouds? If so by approximately by how much? Although only a few days were analyzed, this should be sufficient to make some more definitive statements. The purpose of the abstract is to give the reader a more quantitative summary of the findings and impact of the new results.

2. The paper is quite long, but the justification for the four changes in the DARDAR product are not discussed, instead, there is a list of references. Since these changes are of vital importance, a couple of sentences in each case summarising the evidence would be helpful to the reader. For example, on page 6, line 18, four references are quoted to justify reducing the max value of S (the lidar ratio) from 120 sr to 50 sr, and hence changing the coefficient alpha (lnS) by a factor of three from 0.0237 to 0.008 (page 7,line 9). What sort of observations were used? Were they Raman or HSRL lidar – ground-based or airborne? How comprehensive? How confident are we of any implied change in the radiative properties of thin cold ice clouds?

3. Figure 2 shows the change in the PSD. It would seem that this is crucial to the increase in the IWC, because the longer tail of larger particles with the normalised size above 2.8, will lead to large changes in Z, but smaller changes in IWC, hence a given Z will now correspond to a lower IWC. Is this effect dominant, or is the change in m-d of equal importance? Is the reduction in the concentration of particles with normalised

size below 0.2 of any significance? It would help the reader if these aspects were discussed.

4. The figures are of very poor quality and are scarcely legible.

5. Finally, there are quite a few typos.

---

## Author Comment (AC1) · 2 Apr 2019

We would like to express our thanks to the reviewer for his/her help in improving the paper. We are grateful for the time spent on this review. In what follows, we respond point-by-point to the comments made.

Major issues

1. In the abstract, the impacts of changes are presented in an inconsistent way. The impact for IWC is provided with the upper range while the range for size is given. Up

to 50% for IWC is a huge difference. It will be better that the impacts are accessed in term of mean changes and ranges.

Response: Below is the new abstract.

Change in manuscript: Abstract: In this paper we present the latest refinements brought to the DARDAR-CLOUD product, which contains ice cloud microphysical properties retrieved from the cloud radar and lidar measurements from the A-Train mission. Based on a large dataset of in-situ ice cloud measurements collected during several campaigns performed between 2000 and 2007 in different regions of the globe, the parameterizations used in the microphysical model of the algorithm – i.e. the normalized particle size distribution, the mass-size relationship, and the parameterization of the a priori of the normalized number concentration as a function of temperature – were assessed and refined to better fit the measurements, keeping the same formalism as proposed in DARDAR basis papers. Additionally, in regions where lidar measurements are available, the lidar ratio retrieved for ice clouds is shown to be well constrained by lidar-radar combination or molecular signal detected below thin semi-transparent cirrus. Using this information, the parameterization of the lidar ratio was also refined, and the new retrieval equals on average 35 sr +/- 10 sr in the temperature range between -60°C and -20°C. The impact of those changes on the retrieved ice cloud properties is presented in terms of IWC and effective radius. Overall, IWC values from the new DARDAR-CLOUD product are in average 20% smaller than the previous version. In parallel, the retrieved effective radii increase between 5% and 40%, depending on temperature and the availability of the instruments, with an average difference of +20%. Modifications of the microphysical model strongly affect the ice water content retrievals with differences that were found to range from -50% to +40%, depending on temperature and the availability of the instruments. Larger IWC values are found with the new version in the cold regions detected by the lidar. On the contrary, in warmer regions, where only the radar measurement is available, a reduction of the retrieved IWC is found. The largest differences are found for the warmest temperatures (between

-20°C and 0°C) in regions where the cloud microphysical processes are more complex and where the retrieval is almost exclusively based on radar-only measurements. The new lidar ratio values lead to a reduction of IWC at cold temperatures, the difference between the two versions increasing from 0% at -30°C to 70% below -80°C. Effective radii are not impacted. At cold temperatures, the impact of the new lidar ratio on the retrieved IWC is larger than that of the new microphysical model, hence a reduction of IWC values for the new DARDAR-CLOUD product, for all temperatures.

2. The quality of the figures is poor. The x-axis and y-axis titles are too small, and units also need to be correctly displayed.

Response: This has been modified, an example is presented in Fig. 1.

3. It is not clear why short periods of data were selected for comparison. It will be great that results on seasonal and global scales are presented to document the impacts due to algorithm changes.

Response: This short period of data was used to reduce calculation time. Since Cloud-Sat and CALIPSO have a polar orbit, we considered this subset was enough to test the new version of the algorithm and statistically represent the entire range of the retrieved properties and the impact of a modification of the algorithm, globally.

4. The algorithm outputs lidar ratio, which has limited information from observations directly. The V1 lidar ratio a prior was clearly wrong. The V2 results are more reasonable. However, the results of lidar ratio should be compared with CALIPSO results. Also, it is important to discuss on how multiple scattering is treated in the algorithm because it is directly tied to effective lidar ratio selection.

Response 1: A comparison with CALIPSO can be made, based on the work by Garnier et al. 2015. To do so, only pixels corresponding to single layer semi-transparent cirrus clouds were selected from the dataset, and the statistic of the corresponding retrieved lidar ratios are compared to the parameterization of S as a function of temperature determined by Garnier et al, 2015. This is presented in Fig.2, with the parameterization by Garnier et al in red triangles and the parameterization for the Varcloud algorithm in black solid line with circles. As can be seen, the slope of the parameterization differs from one algorithm to the other, and the difference between the two algorithm increases when the temperature decreases. This is already discussed in the manuscript in section 3.1. Since the retrieved lidar ratio strongly depends on the a priori in the case of DARDAR-CLOUD, it also moves away from the CALIPSO retrieval when the temperature gets colder. However, it is visible that the new version of DARDAR-CLOUD is more in agreement with CALIPSO results. This result is hinted at in the current manuscript by combining the results described by Figure 1 with those presented in Figure 6. Therefore, since the paper is already rather long, we propose not to make any change regarding this matter.

Response 2: Multiple scattering is accounted for in the lidar backscatter forward model. This forward model was developed by Hogan (2008). It uses a fast, approximate analytical method based on the representation of the photon distributions by their variance and covariance to infer multiple scattering effect at each gate of the measured profile.

Hogan, R. J., 2008: Fast Lidar and Radar Multiple–Scattering Models. Part I: Small–Angle Scattering Using the Photon Variance–Covariance Method. J. Atmos. Sci., 65 (12), 3621–3635, doi:10.1175/2008JAS2642.1.

Eloranta, E. W., 1998: A practical model for the calculation of multiply scattered lidar returns. Appl. Opt., 37, 2464–2472.

Change in manuscript (Page 7, line 6): Additionally, multiple scattering is not accounted for the same way. Based on the work from Platt (1973, 2002), Garnier et al (2015) define a multiple scattering factor to correct the two-way transmittance from the contribution of multiple scattering. This correction factor equals 1 in the single-scattering limit and varies from 0.5 to 0.8 as a function of temperature for the CALIOP instrument. In the Varcloud algorithm, multiple scattering is accounted for in the lidar backscatter

forward model that was developed by Hogan (2008). This forward model uses a fast, approximate analytical method based on the representation of the photon distributions by their variance and covariance to infer multiple scattering effect at each gate of the measured profile.

Minor issues

1. Page 3, line 23: change "restate" to "summarize".

Response: This has been changed accordingly.

Change in manuscript (Page 3, line 23): We summarize here the main characteristics of the inverse method. . .

2. Page 4, line 3: change "apparent" to "attenuated".

Response: This has been changed accordingly.

Change in manuscript (Page 4, line 3): Then, the lidar attenuated backscatter at range r from the instrument can be expressed in the single-scattering limit as a function of $\alpha$v and S

3. Page 4, line 11-14: the equivalent diameter discussion here is confusion.

Response: OK, the discussion has been modified, getting more into details.

Change in manuscript (Page 4, line 11-13): D_eq is the equivalent diameter of the ice crystal (in meters). It corresponds to the diameter the particle would have if it was a spherical liquid particle of the same mass. It is obtained from the mass of the particle M and the density of water rho_w=1000 kg/m3 as follows: D_eq=[6M/($\pi$ rho_w )]ˆ(1/3) In the normalized framework, this equivalent diameter is scaled by the mean volume diameter, D_m, defined as the ratio of the 4th and the 3th moments of the PSD, in terms of D_eq.

4. Page 4, line 24: "the LUT" refers which LUT. Is there only one LUT for the algorithm?

Response: Yes, there is only one LUT. This LUT is defined for bins of D_m. Each value of D_m corresponds to a value of $\alpha$/N_0*, Z/N_0*, IWC/N_0*, r_e etc... It is then possible to find an unambiguous relationship between $\alpha$/N_0* and Z/N_0* (for the forward model of radar reflectivity) or between $\alpha$/N_0* and IWC/N_0* (for the retrieval of IWC).

Change in manuscript (Page 4, line 24): Once the optimized cloud profile has been determined, this same LUT is also needed to retrieve additional features of the profile such as the IWC and effective radius.

5. "Page 5, line 4-5: The logic does not make sense. Maintaining continuity does not mean accurate results.

Response: Mass-size relationships are difficult to parameterize since they depend on many factors, and not only temperature. Ideally, each cloud type (and associated cloud processes) should be given its specific parameterization. There have been many studies, some of them are referred to in this paper (and work still ongoing), trying to refine those relationships that are crucial to retrieve cloud properties from remote sensing measurements. However, in the case of the DARDAR-CLOUD product, since we lack information to be able to accurately fit to each and every cloud situation, we decided to focus on the statistical side of the problem. Hence the choice to use a single relationship, that was determined using a selection of cloud in-situ measurements performed on different kinds of ice clouds in different parts of the globe.

Change in manuscript (Page 5, line 4-6): However, temperature is not the only parameter that matters for the determination of M(D). In order to accurately fit this relationship to each and every cloud situation, we would need more information on cloud type, particle size, that are not straightforward to derive from the CloudSat-CALIPSO synergy. In addition, it is difficult to change M(D) in the retrieval scheme upon the cloud type and the meteorological conditions without risking to bring discontinuity on the retrievals. As a result, in the case of the DARDAR-CLOUD product, we decided to focus on statistical results and assume a single M(D) relationship which can work for most of the situations.

6. Page 13, lines 9-10: The two extremes of radar-only results seem indicating that the algorithm for the radar-only region is not very stable

Response: Yes, unfortunately, it is more or less the case. The reason is also that in such regions, the complexity of microphysical processes increases with different impacts of the changes in the algorithm depending on the processes at stake.

7. Table 1: use the formal way to represent the constant coefficients.

Response: Sorry, but we do not understand what you mean by "the formal way".

8. Figure 5a and 5b: It is hard to separate the two line styles. How about using color lines for them?

Response: This has been modified, the new figure is presented Fig. 3.

9. Figure 9, figure caption: What does "33.2" mean here?

Response: It is a mistake. It refers to the paragraph 3.2 "The microphysical model".

Change in manuscript (caption Figure 9): Comparison of the retrieved IWC for the 2 microphysical parameterizations presented in section 3.2 (but with the same lidar ratio a priori).
* * *
[Figure]

[Figure]

Fig. 1.

[Figure]

Semi-transparent single-layer cirrus clouds

Fig. 2.

[Figure]

Fig. 3.

---

## Author Comment (AC2) · 2 Apr 2019

We would like to express our thanks to the reviewer for his/her help in improving the paper. We are grateful for the time spent on this review. In what follows, we respond point-by-point to the comments made.

1. The abstract should be more informative and provide a more precise summary of the findings. At present the statements are too vague. Currently it says IWC can be 'up to 50% with, globally, a reduction'. 50% is a large change. What is the global

average reduction? Effective radius increases between 5% and 40%, with the largest difference in clouds between -20C and 0C. The new lidar ratio of 35 +/10 sr for cold clouds is quite a reduction on the previous values. Line one of the introduction stresses the importance of ice clouds on the radiation budget, but this aspect does not seem to be directly addressed in the rest of the paper. Do changes in effective radius for the warmer ice clouds lead to changes in the radiation budget? Perhaps not, as such clouds are already optically thick? Do changes in the lidar ration affect the radiative properties of the thin cold ice clouds? If so by approximately by how much? Although only a few days were analyzed, this should be sufficient to make some more definitive statements. The purpose of the abstract is to give the reader a more quantitative summary of the findings and impact of the new results.

Response: Regarding the impact of changes in the effective radius and lidar ratio on the cloud radiative properties, it is not the objective of this paper. This paper aims at giving information on the modifications that were made in the algorithm and how the DARDAR-CLOUD product is impacted. As a result, we only focused on variables available in this product (IWC and effective radius of ice clouds).

Change in manuscript: Abstract: In this paper we present the latest refinements brought to the DARDAR-CLOUD product, which contains ice cloud microphysical properties retrieved from the cloud radar and lidar measurements from the A-Train mission. Based on a large dataset of in-situ ice cloud measurements collected during several campaigns performed between 2000 and 2007 in different regions of the globe, the parameterizations used in the microphysical model of the algorithm – i.e. the normalized particle size distribution, the mass-size relationship, and the parameterization of the a priori of the normalized number concentration as a function of temperature – were assessed and refined to better fit the measurements, keeping the same formalism as proposed in DARDAR basis papers. Additionally, in regions where lidar measurements are available, the lidar ratio retrieved for ice clouds is shown to be well constrained by lidar-radar combination or molecular signal detected below thin semi-transparent

cirrus. Using this information, the parameterization of the lidar ratio was also refined, and the new retrieval equals on average 35 sr +/- 10 sr in the temperature range between -60°C and -20°C. The impact of those changes on the retrieved ice cloud properties is presented in terms of IWC and effective radius. Overall, IWC values from the new DARDAR-CLOUD product are in average 20% smaller than the previous version. In parallel, the retrieved effective radii increase between 5% and 40%, depending on temperature and the availability of the instruments, with an average difference of +20%. Modifications of the microphysical model strongly affect the ice water content retrievals with differences that were found to range from -50% to +40%, depending on temperature and the availability of the instruments. Larger IWC values are found with the new version in the cold regions detected by the lidar. On the contrary, in warmer regions, where only the radar measurement is available, a reduction of the retrieved IWC is found. The largest differences are found for the warmest temperatures (between -20°C and 0°C) in regions where the cloud microphysical processes are more complex and where the retrieval is almost exclusively based on radar-only measurements. The new lidar ratio values lead to a reduction of IWC at cold temperatures, the difference between the two versions increasing from 0% at -30°C to 70% below -80°C. Effective radii are not impacted. At cold temperatures, the impact of the new lidar ratio on the retrieved IWC is larger than that of the new microphysical model, hence a reduction of IWC values for the new DARDAR-CLOUD product, for all temperatures.

2. The paper is quite long, but the justification for the four changes in the DARDAR product are not discussed, instead, there is a list of references. Since these changes are of vital importance, a couple of sentences in each case summarising the evidence would be helpful to the reader. For example, on page 6, line 18, four references are quoted to justify reducing the max value of S (the lidar ratio) from 120 sr to 50 sr, and hence changing the coefficient alpha (lnS) by a factor of three from 0.0237 to 0.008 (page 7, line 9). What sort of observations were used? Were they Raman or HSRL lidar – ground-based or airborne? How comprehensive? How confident are we of any implied change in the radiative properties of thin cold ice clouds?
Response: These changes were initiated after the DARDAR-CLOUD product was compared to other satellite products. To account for the differences that were observed, the lidar ratio a priori, the N0' a priori, the normalized PSD and the M-D relationship have been identified as parameters that could be refined, due to the uncertainty and/or questionable reliability of the current parameterizations.

Regarding the changes in the microphysical model, they are justified by the fact that data from new field campaigns with ice clouds in-situ measurements have been made available, providing more accurate measurements of PSDs and IWC and/or a larger statistic of measured ice cloud properties, compared to the data used for the first version of the algorithm. We decided to refine the parameterizations based on this more recent information.

Regarding the references for the lidar ratio, all four studies consist in measurements of thin/semi-transparent cirrus clouds with a simple elastic lidar, using the difference between the backscatter measured above and below the cloud layer to infer the transmission and then the integrated extinction and lidar ratio. Platt et al. (1987) and Garnier et al. (2015) use additional measurements from an infrared radiometer to account for multiple scattering effects. Average cloud temperatures are between -60°C and -40°C and optical depths below 1. The study presented by Platt et al. (1987) is based on measurement of midlatitude cirrus observed with a groundbased instrument located in Aspendale, Victoria (Australia) during one winter and one summer season, as well as observations of tropical cirrus made in Darwin (Australia). Chen et al. (2002) present one year of ground-based lidar measurements in Taiwan. Yorks et al. (2011) compare measurements obtained during five airborne campaigns in different locations in Central and North America and Hawaii. Finally, Garnier et al. (2015) present a statistic of CALIPSO observations over the year 2008. The lidar wavelength is 694nm for the study performed by Platt et al. (1987) and 532nm for the others. Although these measurements are restricted to situations that can be handled with elastic lidars, the average lidar ratio values are in agreement with those obtained using Raman lidars

and presented by Reichardt et al. (2002), Whiteman and Demoz (2004) and Thorsen and Fu (2015). In particular, Thorsen and Fu present a statistic over several years of lidar ratio retrievals using ARM Raman lidars on two locations: Lamont, Oklahoma and Darwin, Australia. This statistic shows lidar ratio values between 5 and 50 steradians, with a maximum of occurrence at 27 sr for Darwin and 22 sr for Lamont.

Finally, about the implied changes in the radiative properties of cold ice clouds, it is again not the objective of this paper.

Change in manuscript (Page 3, line 12): a few issues have been identified. For example, Deng et al. (2013) compared DARDAR-CLOUD with other satellite products and with cloud properties derived from aircraft in-situ measurements obtained with a 2D-S probe, during the SPARTICUS campaign in 2010. Compared to the other CloudSat-CALIPSO product and the aircraft observations, the DARDAR-CLOUD product seemed to overestimate IWC in cloud regions where only lidar measurements were available. Sourdeval et al. (2016) also compared the Ice Water Path (IWP) retrieved with different satellite products over the year 2008 and highlighted the fact that the DARDAR-CLOUD product tends to overestimate IWP, in particular for values below 10 g.m-2.

Change in manuscript (Page 6, line 17-18): This was found to produce values of S that are too large at cold temperatures (up to 120 sr) compared to the climatology. Indeed, several studies on semi-transparent cirrus clouds were performed with elastic lidars in the visible, either from airborne (Yorks et al., 2011), groundbased (Platt, 1987, Chen et al. 2002) or spaceborne (Garnier et al., 2015) instruments. In all cases, retrieved lidar ratios were found around an average value of 25-30 sr and rarely exceeded 50 sr. In addition, more studies were made on cloud optical properties, including measurements performed in the UV by Raman ground-based lidars, showing similar values for the retrieved lidar ratios (Whiteman and Demoz, 2004, Thorsen and Fu, 2015).

Change in manuscript (Page 7, line 21): The idea here is to assess and refine these parameterizations, using a more comprehensive and accurate dataset of ice cloud in-

situ measurements.

3. Figure 2 shows the change in the PSD. It would seem that this is crucial to the increase in the IWC, because the longer tail of larger particles with the normalised size above 2.8, will lead to large changes in Z, but smaller changes in IWC, hence a given Z will now correspond to a lower IWC. Is this effect dominant, or is the change in m-d of equal importance? Is the reduction in the concentration of particles with normalised size below 0.2 of any significance? It would help the reader if these aspects were discussed.

Response: Due to normalization, particles with the normalized size above 2.8 does not only mean large particles but also large particles with respect to the mean size of the distribution (D_m). Those particles correspond to the tail of the distribution. Unless the un-normalized particle size distribution is very broad, these particles have very little contribution to the overall size distribution. As presented by Delanoë et al (2014), the majority of the data is concentrated in the area where D_eq/D_m=1. As a result, the change in M(D) is expected to be of more importance. The same reasoning also applies for normalized sizes below 0.2.

Change in manuscript (Page 8, line 12): However, as presented by Delanoë et al (2014), the majority of the data is concentrated in the area where D_eq/D_m=1. The change in M(D) is therefore expected to be of more importance than the modification of the normalized particle size distribution.

4. The figures are of very poor quality and are scarcely legible.

Response: This has been modified, examples are presented in Fig. 1 and 2.

5. Finally, there are quite a few typos.
* * *
[Figure]

**Fig. 1.**

[Figure]

**Fig. 2.**

---

## Author Response (AR2)

**Response to Associate Editor's comments**

Dear Sir, thank you for your help in improving the paper. We hope that the propositions we make to your comments will satisfy you. Please, refer to the marked-up manuscript for the modifications that were made according to what follows.

**Main comments:**

**1.** Just keeping track of which version that is discussed needs an alert brain. On page 3 it is declared that v2.1.1 will be denoted as v1, but still v2.2.1 is used in the text (e.g. page 7,line1) and it is also called the "old" (e.g. in text of Figure 1). In addition, it has already caused us confusion to call the "old" version v1 in the manuscript, when it has the official name v.2.1.1. Do you know the official version number of the "new" version? If it will v3.x, I strongly recommend to use call the old version v2 and the new v3. Anyhow, consider the naming and how the names are used in the text.

**Response**: I can understand that this naming can be confusing. We keep this V1 / V2 naming, removing any possible renaming, such as v2.1.1 (it was a mistake).

**2.** As a side note, the ATBD found at the ICARE site

http://www.icare.univ-lille1.fr//projects_data/dardar/docs/varcloud-algorithm_description-v1.0.pdf

has a name indicating that is valid for v1. Is it also valid for v2.1? I suggest to clarify this on the site.

**Response**: The ATBD on the ICARE website is not up-to-date: it still mentions a parametrization as a function of height for the extinction-to-backscatter ratio, and this only when radiances are assimilated. For version 2.1.1 of DARDAR-CLOUD, however, this was changed for a parameterization as a function of temperature. This will be clarified on the site, and a new ATBD will be produced.

**3.** I see a logic in have you organised Sections 2 and 3, but I think it causes problems. Take the mass-size relationship as example. It is introduced on page 5 and you have quite a bit of discussion here. You get back to the subject in the start of Sec 3.2, and have another relative long discussion. Here you introduce the "BEST" parameterisation, and as a reader you then think that this is the one that you have selected (as you call it BEST). But still no values given. The story is continued in Sec 3.2.3, and first here the final parameterisation is mentioned ("Composite"). And then you have to go to Table 1 to actually see the values selected. Note also that you define BFM on page 5, but you still go back and use citations here and there.

The story around x and y of Eq. 5 is shorter but the selected values are only reported as comments inside a figure. If you are looking for these values specifically, it will probably take

a while before you locate the values. And here are the old values not reported at all, which should be given as well.

There are more similar examples, but I hope you get my point without listing those as well.

My suggestion is to clearly report all parameter values of concern in tables (both old and new ones, following the present Table 1), and refer to the tables already when the parameters are introduced in Sec 2. The reader can then easily find and compare old and new parameter values, without risk of confusion. How you selected the new values can still be discussed in Sec 3. I would also suggest to make Sec 2 as condensed as possible and to avoid discussing the same subject both in Sec 2 and 3.

**Response**: Sections 2 and 3 have been completely rearranged and tables with old and new parameters added. The "BEST" parameterization refers to a set of mass-size relationships, each one linked to a specific measurement campaign. It was called "BEST" because each campaign was assigned the M-D relationship allowing for the best match between measured bulk IWC and IWC retrieved using M-D. But since this naming seems confusing, it was changed to "RETRIEVED". The process of assigning a specific M-D to each campaign is explained in Delanoë et al. 2014 as well as the detailed expressions of each M-D and therefore we did not want to describe them again (reference to Table 3 in Delanoë et al. 2014 is given).

**4.** I hesitate to suggest things that makes the manuscript longer, and even more hesitation to bring up my own work. So take the following lines only as a tentative suggestion, originating from my personal curiosity. DARDAR IWP has been used as broad reference for the treatment of ice in models and products. The most recent one is from us: https://www.atmos-chem-phys.net/18/11205/2018/. It seems that the new version gives lower mean IWP. Possible to give a rough number?

**Response**: The following figure gives a summary of the impact of the new parameterizations on IWC and IWP. Smaller values of IWC tend to lead to smaller IWP, of about 24%. This information is briefly added in the conclusion of the paper.

[Figure]

**Further comments:**

**5.** The abstract needed to be changed, but has now become quite long. It can be shortened by removing less important comments, such as "collected during several campaigns performed between 2000 and 2007 in different regions of the globe". In addition, the sentences "Larger IWC values are found with the new version in the cold regions detected by the lidar." and "The new lidar ratio values lead to a reduction of IWC at cold temperatures" seem to contradict.

**Response**: Please, have a look at the new abstract in the marked-up version of the manuscript.

**6.** Page 1, line1-3: For me the sentence does not make sense. Anyhow, I suggest to remove the sentence as it is not needed. The motivation should be known to readers of AMT.

**Response**: This sentence has been removed.

**7.** Page 4, line 11: Another example where the information is spread out. As a reader you get the impression that a full representation of the S profile is included in the state vector, while below Eq. 10 it becomes clear that the state vector only contains a and b. More generally, the way you discuss the state vector also raises the question if the formulation of x has changed or not? If there is no change, why not refer to the article where x is best presented? That is, either skip the details, or clearly state initially that it is a recap (and keep all discussion of the content of x together).

**Response**: With the modifications of Sections 2 and 3, hopefully this problem has been solved.

**8.** Page 5, line 7: Is Z calculated for totally random orientation or horizontally aligned particles? (I have failed to find a clear statement of this in the DARDAR documentation, so would be good to get it clarified.)

**Response**: Z is calculated for totally random orientation.

**9.** Page 5, line 24: Not just "especially", it will not work without a priori.

**10.** Page 8, line 33: As far as I can see alpha and beta are not defined inside the manuscript. This is an important change and should be visible in a table. Also note the risk of confusion between this alpha and visible extinction.

**Response**: alpha and beta are briefly introduced in section 2, along with the modified gamma shape. A table giving the values of these parameters for the two versions has been added and they are now called alpha_F and beta_F.

**11.** Page 12, line 4: Don't understand "a small error was set on the a priori in the cost function". Do you mean that the a priori uncertainty applied is small?

**Response**: Yes, that is right.

**12.** Page 12, line7-14: This part took me a while to get. Consider a rewrite.

**Response**: Please, refer to the marked-up manuscript for the modifications. The explanation has been simplified.

**13.** Page 13, line 1: "2" should be "two". This applies also to other places in the manuscript.

**14.** Page 17: Some error here. Most references are written in italic.

**15.** Figure 3b and 3c: Should it not be ln(N') and ln(N*0) on the x-axis.

**Response**: It should be ln(N') for Figure 3b, indeed.
As for panel 3c, it is $N*0 = \exp(xT + y)/\alpha^b$ that is presented (in log scale), for different values of alpha.

**16.** Figure 4: should not the last reference be Heymsfield et al (2014).

**Response**: Indeed, it was a mistake, it should rather be Heymsfield et al (2010).

**17.** Figure 9 and 10: Top row of panels are identical to Fig 8, so seems unnecessary to repeat these panels. And just good to make the figures smaller.

[revised manuscript text omitted]

---

## Author Response (AR3)

I would just like to make a precision here on the first comment made by Mr. Eriksson about the different versions of the DARDAR-CLOUD product:

"**1.** Just keeping track of which version that is discussed needs an alert brain. On page 3 it is declared that v2.1.1 will be denoted as v1, but still v2.2.1 is used in the text (e.g. page 7,line1) and it is also called the "old" (e.g. in text of Figure 1). In addition, it has already caused us confusion to call the "old" version v1 in the manuscript, when it has the official name v.2.1.1. Do you know the official version number of the "new" version? If it will v3.x, I strongly recommend to use call the old version v2 and the new v3. Anyhow, consider the naming and how the names are used in the text.

**Response**: I can understand that this naming can be confusing. We keep this V1 / V2 naming, removing any possible renaming, such as v2.1.1 (it was a mistake)."

**New response:** The answer from the operational team at the ICARE centre came only after the response was sent to AMT and contrary to what was said beforehand, it was decided to call the new version V3. As a result, changes have been made in the manuscript, calling the old version (DARDAR-CLOUD v2.1.1) V2 and the new one V3. Apologies for the late reply and confusion.